# Regulation of Oxidative Phosphorylation of Liver Mitochondria in Sepsis

**DOI:** 10.3390/cells11101598

**Published:** 2022-05-10

**Authors:** Pierre Eyenga, Benjamin Rey, Lilia Eyenga, Shey-Shing Sheu

**Affiliations:** 1Center for Translational Medicine, Department of Medicine, Thomas Jefferson University, Philadelphia, PA 19107, USA; shey-shing.sheu@jefferson.edu; 2Laboratoire de Biométrie et de Biologie Evolutive, CNRS, Université Claude Bernard Lyon 1, UMR5558, F-69622 Villeurbanne, France; benjamin.rey@univ-lyon1.fr; 3Faculté de Pharmacie de Lyon, ISPB, Université Claude Bernard Lyon 1, 8 Avenue Rockefeller, F-69008 Lyon, France; liliaeyenga@yahoo.fr

**Keywords:** liver, sepsis, mitochondria, oxidative phosphorylation, cytochrome c oxidase, ATP synthase

## Abstract

The link between liver dysfunction and decreased mitochondrial oxidative phosphorylation in sepsis has been clearly established in experimental models. Energy transduction is plastic: the efficiency of mitochondrial coupling collapses in the early stage of sepsis but is expected to increase during the recovery phases of sepsis. Among the mechanisms regulating the coupling efficiency of hepatic mitochondria, the slipping reactions at the cytochrome oxidase and ATP synthase seem to be a determining element, whereas other regulatory mechanisms such as those involving proton leakage across the mitochondrial membrane have not yet been formally proven in the context of sepsis. If the dysfunction of hepatic mitochondria is related to impaired cytochrome c oxidase and ATP synthase functions, we need to consider therapeutic avenues to restore their activities for recovery from sepsis. In this review, we discussed previous findings regarding the regulatory mechanism involved in changes in the oxidative phosphorylation of liver mitochondria in sepsis, and propose therapeutic avenues to improve the functions of cytochrome c oxidase and ATP synthase in sepsis.

## 1. Introduction

Sepsis and septic shock remain a main cause of death in intensive care units despite numerous therapeutic trials [1]. Sepsis-related mortality frequently results from multi-organ failure, characterized by impaired pulmonary function, liver failure, cardiac dysfunction, acute renal failure and disseminated intravascular coagulation [2].

Liver failure associated with sepsis is often considered a late-stage feature of critical illness, manifesting as jaundice and hyperbilirubinemia [3]. However, recent experimental studies have revealed that liver dysfunction is an early event in sepsis [4,5]. Early depression of hepatocellular function is characterized by the decreased capacity of the liver mitochondria to supply adenosine triphosphate (ATP) [6]. This is associated with a higher rate of reactive oxygen species generation by liver mitochondria [6], the accumulation of oxidative damage and the clearing of dysfunctional mitochondria (i.e., mitophagy) [7]. Although these early processes lead to downregulation of hepatocellular function, they do not appear to result in cell death or irreversible structural damage [8]. It has been suggested that this initial loss of function and decrease in aerobic respiration may be a protective adaptative response that prevents cell death, allowing function to return over time [9]. It has therefore been predicted that late recovery of liver function is associated with increased efficiency of mitochondrial oxidative phosphorylation [10] and/or activation of mitochondria biogenesis [11].

In a previous study, we tested this hypothesis by examining the time course of the functional properties of liver mitochondria in a rodent model of severe sepsis [12]. Contrary to expectations, we found that restoration of efficient mitochondrial oxidative phosphorylation did not occur 36 h after the onset of sepsis, despite the induction of mitochondrial biogenesis mechanisms [12]. From a mechanistic point of view, the lack of an improvement in mitochondrial function seems to be related to a marked decrease in the content of cytochrome c oxidase and ATP synthase and their impact on oxidative phosphorylation. The aim of this article was to describe the changes in oxidative phosphorylation efficiency that occur in liver mitochondria during sepsis, to explore the different regulatory mechanisms involved and to clarify the role of cytochrome c oxidase in this context. We also discuss some therapeutic avenues that could improve the functions of cytochrome c oxidase and ATP synthase and restore mitochondrial oxidative phosphorylation efficiency in patients with sepsis.

## 2. Mitochondrial Oxidative Phosphorylation

### 2.1. Mitochondrial Structure

Mitochondria are double-membrane intra-cytoplasmic organelles found in most eukaryotic cells, with the exception of mammalian erythrocytes. The outer membrane enclosing the organelle forms a semi-permeable barrier separating the mitochondria from the cytosol. The outer mitochondrial membrane is freely permeable to small molecules and contains special channels capable of transporting large molecules. The inner membrane delimits two operating volumes: the matrix and the intermembrane space between the inner and outer membrane (Figure 1). The inner membrane has many cristae containing the protein components of the electron transport chain (ETC), which is composed of transmembrane protein complexes (Complexes I to IV) and the freely mobile electron transfer carriers ubiquinone and cytochrome c. The ETC complex is organized as a super-complex specifically configured to function properly [13]. These assembled components, together with F1F0-ATP synthase (Complex V), form the basis of ATP production during oxidative phosphorylation (Figure 1). Oxidative phosphorylation is defined as the phosphorylation of adenosine diphosphate (ADP) to ATP using energy stored as a proton electrochemical gradient derived from the ETC’s activity. During this process, the ETC uses the oxidation of reducing equivalents from metabolic substrates to transfer protons across the inner mitochondrial membrane via three proton pumping enzyme complexes (NADH: ubiquinone oxidoreductase (I), cytochrome c reductase (III) and cytochrome c oxidase (IV)). The accumulation of protons in the intermembrane space creates an electrochemical gradient known as a proton motive force, which is used by the ATP synthase to catalyze the endergonic synthesis of ATP from ADP and inorganic phosphate. ATP generated in the mitochondrial matrix is then exported to the cytoplasm by the adenine nucleotide translocator (ANT) to fulfill the cytoplasmic energy demand.

### 2.2. Oxidative Phosphorylation Efficiency

Mitochondrial oxidative phosphorylation is a major source of hepatocellular ATP production under aerobic conditions. The energy demand of the cell is difficult to quantify during sepsis. Nevertheless, the ability of mitochondria to sense and respond to changes in nutrient availability and energy demand is critical for the maintenance of hepatocellular homeostasis. Within the mitochondria, the coupling between the rate of oxidation and the rate of ATP generation (phosphorylation) is flexible and can be regulated [14]. In a state of homeostasis, the efficiency of oxidative phosphorylation is set such that the coupling between substrate oxidation (mitochondrial respiration) and ATP production can meet the cellular demand for ATP. Conversely, when the stimulation of substrate oxidation (mitochondrial oxygen consumption) does not lead to an increase in ATP generation in stoichiometric proportion, this expresses an uncoupling of the oxidation and phosphorylation reactions. Therefore, studying the oxidative phosphorylation efficiency of liver mitochondria isolated from septic animals provides information on the ability of mitochondria to meet hepatocellular ATP demand.

#### 2.2.1. Quantification of Mitochondrial Efficiency 

In a complex metabolic network such as mitochondrial oxidative phosphorylation, where the coupling between respiration and adenosine diphosphate (ADP) phosphorylation can be regulated at different levels, a rigorous quantitative analysis measuring the rate of ATP synthesis associated with oxygen (O_2_) consumption allows the overall efficiency of the reactions (ATP:O) to be determined.

Two methods are generally described in the literature for assessing mitochondrial oxidative phosphorylation efficiency in vitro. An indirect approach is to measure the ADP:O ratio, defined as the amount of O_2_ required to allow the phosphorylation of a known amount of ADP. With this method, the yield of oxidative phosphorylation is estimated from the mitochondrial oxygen consumption in a fully phosphorylating active state (State 3: a high concentration of ADP). This method is commonly used but has the major drawback that the ATP generated from exogenous ADP is only measured at a single level of energy demand: the maximum State 3 phosphorylation rate (State 3), which is far from the physiological state. Another approach is to directly quantify mitochondrial ATP production, either by using a magnesium probe that binds to ATP and ADP [15] or by measuring ATP production spectrophotometrically [6,12,16,17] in a system using hexokinase (HK) and glucose-6-phosphate dehydrogenase (G6-PDH) to convert ATP to nicotinamide adenine dinucleotide phosphate (NADPH) (Figure 2A). No assumption about the amount of ATP produced is required in this approach, as ATP production is quantified based on the linear relationship with NADPH absorbance. Furthermore, since hexokinase hydrolyzes ATP as it leaves the mitochondria, the ADP and ATP concentrations remain stable (i.e., “clamped”) throughout the experimental period [18], allowing quantification of oxidative phosphorylation under steady-state rates. In the context of sepsis, quantification of the efficiency of liver mitochondria using ADP:O methods yielded conflicting results, although the authors used a peritonitis model of sepsis such as cecal ligation and puncture (CLP) that mimicked human sepsis [19,20,21]. Using this gold standard model of experimental sepsis [22], Herminghaus et al. [19] showed a time-dependent decrease in ADP:O (i.e., 96 h after the induction of peritonitis), while other authors showed an increase in the oxygen consumption of isolated liver mitochondrial [20] and no change in ADP:O [21]. These results [20,21] were surprising, given that the oxygen consumption of hepatocytes was found to be lowered during sepsis [20] and that a depletion of hepatocellular ATP had previously been observed [23] using the same experimental model. Moreover, several years later, Brealey and collaborators confirmed, in a clinical study, the depletion of ATP in the muscles of a septic patients and linked this depletion to the poor outcome [24]. The question of when the ATP depletion occurs during sepsis and the mechanisms involved thus remained to be explored. Clearly, a rebound increase in oxygen consumption and ATP production in the recovery phase of sepsis would reflect mitochondrial recovery. In recent studies with a rodent model of peritonitis, we used the second approach to study the time course of oxidative phosphorylation in liver mitochondria during sepsis [6,12]. This protocol allows the measurement of ATP generation and oxygen consumption while maintaining constant low concentrations of ADP in the respiratory medium to create phosphorylating conditions close to those of intact cells where the oxidative phosphorylation state is between States 4 and 3 [25]. The ATP:O ratio can be read, even in the presence of some intrinsic uncoupling, from the slope of the linear relationship between phosphorylation and the respiratory rate (Figure 2C) [26]. Using this approach, we established a time course of liver mitochondrial function by measuring the effective ATP production related to oxygen consumption of mitochondria isolated from the livers of septic rats at different time points during the development of sepsis (i.e., 6, 24 and 36 h). In these studies [6,12], we showed that sepsis leads to a change in ATP turnover in liver mitochondria, with the linear relationship between ATP production and respiration shifting to the right, indicating some degree of uncoupling between ATP generation and oxygen consumption by the mitochondria [12]. Such a decrease in energy efficiency (uncoupling of oxidative phosphorylation) reflects an alteration of the processes involved in the regulation of mitochondrial oxidative phosphorylation caused by sepsis. Three main non-exclusive mechanisms may be involved, either (1) at the level of membrane conductance (proton and or cation leakage), (2) at the coupling site between the electron and proton fluxes through the respiratory chain, or (3) at the level of coupling between the proton and ATP synthesis fluxes through mitochondrial ATP synthase [14]. Note that there are other mechanisms for regulating mitochondrial ATP synthesis, such as depletion of the substrate for oxidative phosphorylation [27,28] or the dissipation of mitochondrial membrane potential through the mitochondrial permeability transition pore (mPTP), which have already been extensively reviewed [29,30]. Nevertheless, we focused on the three main mechanisms of regulation of oxidative phosphorylation efficiency mentioned above and deciphered those that are relevant in the context of sepsis as summarized in Figure 3. 

#### 2.2.2. Regulation of Mitochondrial Oxidative Phosphorylation in Sepsis

##### A. Proton Conductance and Proton Leakage through the Inner Membrane

The chemiosmotic theory proposed by Peter Mitchell in 1961 states that substrate oxidation and ATP synthesis are indirectly coupled through the establishment of an electrochemical proton gradient or proton motive force Δp across the inner mitochondrial membrane. This force, which act on protons, comprises an electrical component (membrane potential, ΔΨ_M_) and a chemical (pH gradient, ΔpH) component [31], and implies that the inner membrane has low intrinsic conductance to protons. However, isolated liver mitochondria energized with NADH or FADH_2_ exhibit a basal oxygen consumption level even if no ADP is provided in the respiratory medium (Figure 2B). This mitochondrial respiration in the absence of ATP synthesis (non-phosphorylating State 2) compensates for the loss of protons through the inner mitochondrial membrane. The addition of ADP and Pi stimulates ATP synthesis (phosphorylating State 3) through the return of protons through ATP synthase, decreasing the proton motive force and thus stimulating respiration. Inhibitors such as oligomycin, which blocks the proton channel of ATP synthase, thus inhibit the respiration related to phosphorylation processes (non-phosphorylating State 4). The oligomycin-insensitive respiration is partly explained by a passive return of protons to the matrix across the inner mitochondrial membrane and provides information on the membrane’s conductance to protons. This process of membrane leakage constitutes a shortfall in ATP synthesis through a decrease in the proton motive force. Although an increase in non-phosphorylating State 4 respiration in hepatic mitochondria is observed during sepsis [12,32], it does not appear to be related to an increase in proton leakage, as explained below. In studies of proton leakage in vitro from isolated liver mitochondria, proton leakage-associated respiration could be measured as a function of the membrane potential (ΔΨ_M_) and provided information on membrane conductance to proton [33]. Figure 1 (insert) shows a typical graph of the kinetic response of membrane proton leakage (oligomycin-insensitive respiration) to membrane potential in isolated liver mitochondria. Whereas Ohm’s law would predict that the intensity of proton leakage through the inner membrane should increase linearly as the membrane’s potential increases, the relationship between proton leakage and membrane potential is not linear at high membrane potential values. In other words, the proton leakage does not depend linearly on its driving force, indicating that proton conductance increases greatly at high potentials [34]. We concomitantly studied mitochondrial proton leakage and oxidative phosphorylation efficiency [6,12] in hepatic mitochondria and found that the non-ohmic part of the curve in septic rats was very similar to that of the controls, although the mitochondrial oxidative phosphorylation efficiency showed uncoupling at 36 h after CLP [12]. These results suggest that the decrease in the efficiency of mitochondrial oxidative phosphorylation observed at the later stages of sepsis (i.e., 36 h after CLP) is not driven by increased membrane conductance to protons. Although it has been shown that protons can return to the matrix through movement at the lipid–membrane protein interface, it has been proposed that under “basal” conditions, only a small percentage of basal leakage is mediated by the lipid bilayer and that the presence of membrane proteins is necessary to increase membrane conductance to protons [35]. During sepsis, a decrease in the mitochondrial cardiolipin concentration would reflect a decrease in the mitochondrial mass rather than disorganization of the inner membrane, which is contrary to the hypothesis of proton leaks at the lipid–membrane protein interface [36]. Taken together, these observations suggest that the involvement of transporters, such as adenine nucleotide transporter (ANT) [37], and/or uncoupling proteins (UCPs) [14] appear to be decisive in the dissipation of the mitochondrial proton gradient during sepsis. The increased expression and activity of these proteins could therefore explain the increase in non-phosphorylating State 4 respiration observed during sepsis [38]. In this regard, UCP2 has been proposed as a potential oxidative stress-induced uncoupling protein in liver mitochondria during endotoxemia [39]. We have recently shown that upregulation of UCP2 mRNA synthesis is accompanied by a decrease in ROS generation in the liver mitochondria of septic rats [12] but does not affect proton leakage. Similar results on proton leak were described by Yu et al. after endotoxemia in mice [40]. Even if UCP2 is capable of transporting protons [41], our previous study showed that UCP2 activation in the liver during sepsis is devolved to ROS dissipation [12,42] but has no measurable impact on mitochondrial proton leakage [12,40]. However, this effect appears to be transient and is unlikely to explain the uncoupling of oxidative phosphorylation efficiency observed with unchanged proton leakage. With regard to ANT, which may also participate in the dissipation of the proton gradient, it has been shown that the amount of this protein is depleted during sepsis [43]. In this context, it is highly unlikely that ANT contributes to the increased leakage of protons and the associated decrease in the proton gradient during late sepsis. We expected that mechanisms other than changes in conductance to protons could be involved in the uncoupling of mitochondrial oxidative phosphorylation described in the late period of sepsis.

##### B. Changes in Proton Pumping Stoichiometry

Originally described by Pietrobon et al. [44], alteration of the proton pumps expresses a change in the amount of protons stored in the intermembrane space (redox slipping) or the amount of protons returning to the matrix via ATP synthase (proton slipping), for the same amount of oxidized electrons.

##### B.1 Redox Slipping

Redox slipping (intrinsic changes) of the ETC involves all compounds or mechanisms leading to a decrease in the intrinsic efficiency of proton pumps. The “redox slipping” or “slippage” of proton pumps within the respiratory chain results in reduced proton pumping for the same number of electrons transferred along the respiratory chain [14]. This reduced efficiency can be balanced by increased respiration to maintain ATP synthesis. This may occur at all three proton pumping sites (Complexes I, III and IV) in response to the interaction of their thiol groups with nitric oxide metabolites [45] and may be favored by the large nitric oxide (NO) release [46] and the high level of NO metabolites observed during sepsis [47]. Studies of the activities of liver mitochondrial complexes in sepsis have shown the unchanged activity of Complex III [47,48], persistent inhibition of Complex IV [47,48] and variable impairment of Complex I [47,48]. Zapelini et al. [48] found no inhibition of Complex I, while Brealey et al. [47] showed a correlation between Complex I inhibition and the severity of sepsis [47]. The discrepancies in the results on the activities of ETC complexes during sepsis led us to measure mitochondrial ATP:O first with specific substrates of Complex I and then with specific substrates of Complex II. Our studies [6,12] showed that the uncoupling of mitochondrial oxidative phosphorylation occurred at 36 h after CLP in rats, irrespective of the substrate used, and confirmed that consistent and early inhibition of Complex IV occurs during sepsis. This observation led us to conclude that only a change in the H+/e- stoichiometry of Complex IV (cytochrome c oxidase) during sepsis appears to have physiological significance in altering the efficiency of mitochondrial oxidative phosphorylation, as observed in other liver diseases [49,50]. Cytochrome c oxidase pumps protons from the matrix to the intermembrane space with two different purposes and through two distinct pathways [51]: (i) the reduction of dioxygen by 4 electrons to form 2 water molecules, where the H+/e– stoichiometry is fixed by the chemical reactions and cannot vary; (ii) the constitution of the proton motive force, where the H+/e– stoichiometry of the pumped protons may vary depending on the available energy of the redox reactions and how the electrons and the proton flux are coupled. It has been proposed that structural modifications at the surface of cytochrome c oxidase [52] or a decrease in its content [12,53,54] or downregulation of mRNA expression of the key components [53,55] could result in electron transfer without proton pumping and thus decrease the overall ATP:O [12,52]. A decrease in cytochrome c oxidase subunit 1 content is observed in several organs during sepsis [12,53,54] and has been linked to downregulation of mRNA synthesis [53,55] and/or protein depletion [56] following its oxidation and nitration [57,58]. NO, which is released in large quantities in the liver during sepsis [46], may have different effects on mitochondrial ATP:O. Short-term incubation of liver mitochondria with NO improves the ATP:O [59]; however, long-term incubation of NO with mitochondrial proteins lead to the formation of peroxynitrite, and inhibition of mitochondrial respiration and ATP synthesis [60]. In addition to the direct interactions of NO metabolites with cytochrome c oxidase that could decrease ATP generation, it has been proposed that cell signaling mechanisms contribute to changes in cytochrome c oxidase’s stoichiometry and ATP generation during sepsis [56]. In this regard, Huttemann et al. [56,61] have shown that the phosphorylation of cytochrome c oxidase triggered by TNFα activation leads to a decrease in ATP generation in sepsis. TNFα is known to be a determining mediator of early sepsis [27,46]. The kinetics of TNFα’s appearance in the liver vary considerably between the experimental models of sepsis used and have led to mixed conclusions [56] regarding the role of cell signaling as a mechanism of regulating oxidative phosphorylation. Instead, we suggest that a global (multifactorial) degradation of mitochondrial proteins and decrease in their activities occurs during sepsis (Figure 3B). As noted by Tavakoli [32], a delay in mitochondrial protein resynthesis may be observed despite the activation of biogenesis [11,12] (Figure 3C). This delay in overall mitochondrial protein resynthesis during sepsis could explained by mitochondrial oxidative phosphorylation uncoupling related to slippage at the cytochrome c oxidase level [32]. This hypothesis is also supported by clinical observations of overall mitochondrial protein degradation in the muscles of septic patients, which contributes to the high incidence of muscle weakness in patients recovering from severe sepsis [62]. In contrast to sepsis, other hepatic pathologies that are accompanied by a decrease in cytochrome content paradoxically show an increase in ATP:O [50]. All of these observations confirm that, through variations in the H+/e- stoichiometry, cytochrome c oxidase appears to be a major site of adjustment of oxidative phosphorylation at the level of the respiratory chain, provided that ATP synthase is fully functional, which does not seem to be the case during sepsis, as we shall see below.

##### B.2 Proton Slipping

This uncoupled proton leakage mechanism occurs at the level of ATP synthase and result in a decrease in the amount of ATP synthesized for a given proton flux. Such an increase in H+/ATP stoichiometry has been demonstrated in vitro [63]. In ATP synthase, the rotational mechanism is not always strictly coupled to proton flux, particularly when nucleotide concentrations are low, or when ATP synthase subunits are depleted [12,64]. In a previous study [12], we showed that ATP:O was unchanged 6 h after CLP and that this was associated with maintenance of the β-subunit content of ATP synthase. Conversely, 36 h after CLP, a severe decrease in the β-subunit of ATP synthase was associated with an uncoupling of oxidase phosphorylation. With regard to the parallel evolution of the ATP synthase β-subunit pool and mitochondrial ATP:O, it appears that in sepsis, a decrease in ATP synthase content leads to ATP production below its maximum [12,64,65]. It should be noted that depletion of ATP synthase content may also decrease the ATP/ADP ratio, which, in turn, should stimulate cytochrome c oxidase and increase ATP production [66]. However, this mechanism is unlikely to occur during sepsis, as our studies [6,12] have shown that the decline in ATP generation does not result in stimulation of cytochrome c oxidase activity. Moreover, it has been shown that a decrease in the ATP subunit content could affect the dimerization/oligomerization of ATP synthase and the formation of invaginations in the inner mitochondrial membrane, which ultimately become mitochondrial cristae [67], with a direct effect on ATP generation [68]. For instance, depletion of the β-subunit could lead to a loss of mitochondrial cristae and super-complex assembly [69,70] and an increase in the H+/ATP ratio without affecting the membrane potential (ΔΨ_M_) [51]. Further studies are needed to establish the extent to which this mechanism plays a role in modulating the efficiency of liver mitochondria during sepsis.

## 3. Therapeutic Implications

Potential therapeutic strategies targeting the mitochondria, including treatment with antioxidants, ROS scavengers and membrane stabilizers (e.g., cyclosporine A, melatonin), have already been extensively reviewed [71,72,73]. However, very few studies have focused specifically on cytochrome c oxidase, which is a major point of the adjustment of mitochondrial function in sepsis. Nevertheless, Deutschman’s group has shown the beneficial effect of exogenous cytochrome c supplementation [74,75], and glutamine [76] and caffeine [77] infusions on cytochrome c oxidase activity and cardiac function in septic mice. Whether such supplementation can improve the efficiency of oxidative phosphorylation and contribute to organ recovery from sepsis [9,10] remains to be explored. Moreover, the authors did not quantify the content of the cytochrome c oxidase subunit, so it cannot be excluded that the increase in oxygen consumption reflected increased oxygen reduction, but without efficient proton pumping [52]. Two recent studies have examined the effect of common symptomatic treatments for sepsis on the function of cytochrome c oxidase in the liver [78] and diaphragm in a rodent model of sepsis [79]. Yang et al. [78] have shown that fluid resuscitation improves the function of cytochrome c oxidase; however, the mechanism remains unclear. Recently, we have shown, in an experimental model of sepsis, that early initiation of mechanical ventilation restores mitochondrial function in the diaphragm. In this study, not only did mechanical ventilation preserve the cytochrome c oxidase content and activity in the diaphragms of septic rats, but it also maintained the efficiency of oxidative phosphorylation (ATP:O) and lowered mitochondrial ROS production by Complex III [79]. In addition to these simple resuscitation measures, interfering with the inflammatory system [80] or cell signaling pathways [56] could improve the function of cytochrome c oxidase. For example, a synthetic cortisol homolog, dexamethasone, has been shown to restore the function of cytochrome c oxidase and prevent kidney injury 24 h after cecal ligation and puncture in mice [80] and to increase the levels of F1Fo ATP synthase and adenine nucleotide translocator (ANT) subunits [81]. However, administration of corticosteroids did not reduce 28-day mortality but increased metabolic disorders in sepsis [82]. The non-receptor tyrosine kinase Src and protein kinase Cε (PKCε) have been identified as positive regulators of cytochrome c oxidase. Treatment with PKC activators, namely diacylglycerol or 4β-phorbol 12-myristate 13-acetate (4β-PMA), resulted in increased cytochrome c oxidase activity in neonatal rat cardiomyocytes [83]. As far as we are aware, these treatments are not yet used to treat sepsis. Other treatments to improve mitochondrial ETC functioning have been proposed. Among them, parenteral injection of succinate could improve ATP concentration in the liver after LPS injection in rats [84] and reduce mitochondrial ROS generation in the kidney after CLP in rats [85]. Although the above mentioned therapeutics may improve mitochondrial ETC function in sepsis, an approach involving the rapid synthesis of respiratory chain proteins deserves consideration. In this regard, sirtuin-3 (SirT3) an NAD^+^-dependent nuclear histone deacetylase [86], and songorine, an active C_20_-diterpene alkaloid from the lateral roots of Aconitum carmichaelii [87], have been shown to induce mitochondrial biogenesis and restore cardiac function in an LPS model of septic myocardiopathy. These molecules represent new therapeutic hopes for countering the pathological effects of sepsis, to be tested in clinical trials.

## 4. Conclusions

The concept of coupling efficiency is essential to explain mitochondrial and liver dysfunction during sepsis and understanding it is necessary for the establishment of new forms of therapy. Depletion and downregulation of cytochrome c oxidase and ATP synthase appear to be consistent and are predominant mechanisms in the occurrence of uncoupling of liver oxidative phosphorylation during sepsis. The mechanism of cytochrome c oxidase depletion could involve phosphorylation of its subunit 1 [61] at an early stage of sepsis through early pro-inflammatory mediators (i.e., TNF α) in addition to its early interaction with NO [45], which leads to a lack of ROS regulation at its level [79], and consequently to oxidation and depletion of mitochondrial proteins, including ATP synthase [12] (Figure 3B). The decrease in cytochrome c oxidase and ATP synthase subunits prevents the changes in stoichiometry of these two enzymes that should take place early in response to sepsis to compensate for the decrease in ATP and the increase in ROS generation. These changes in stoichiometry are an instantaneous response to ETC failure because biogenesis compensation takes much longer to occur in the recovery phase of sepsis (Figure 3C). As a consequence, therapies combining the acceleration of mitochondrial biogenesis and improvement of the stoichiometry of both enzymes should be considered in therapies to restore liver mitochondrial function during sepsis.

## Figures and Tables

**Figure 1 cells-11-01598-f001:**
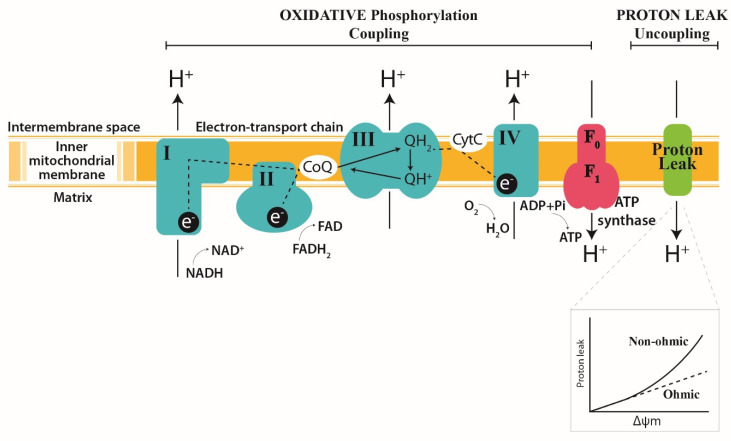
Diagram of the inner mitochondrial membrane, showing three processes that underlie oxidative phosphorylation. (1) The electron transport chain; (2) the uncoupling of oxidation from phosphorylation by leaking H+ across the inner mitochondrial membrane (Leak); and (3) phosphorylation to generate ATP (ATP synthesis). Complex I and II oxidize NADH and FADH_2_ respectively, transferring the resulting electrons to ubiquinol, which carries electrons to Complex III. Complex III shunts the electrons across the intermembrane space to cytochrome c, which brings electrons to Complex IV. Complex IV then uses the electrons to reduce oxygen to water. The energy liberated by the flow of electrons is used by Complexes I, III and IV to pump protons (H+) out of the mitochondrial matrix into the intermembrane space. This proton gradient generates the mitochondrial membrane potential that is coupled to ATP synthesis by Complex V (ATP synthase) from ADP and inorganic phosphate (Pi). Light green and the insert show the proton leak’s kinetics. The respiration rate increases disproportionately with the membrane potential (non-ohmic proton conductance). NADH, reduced nicotinamide adenine dinucleotide; NAD+, oxidized nicotinamide adenine dinucleotide; FADH, reduced flavin adenine dinucleotide; FAD+, oxidized flavin adenine dinucleotide. The dotted line indicates the electron pathways. The green arrows represent substrate reactions. The black arrows show the proton circuit across the IMM. Complexes I-V are marked as I, II, III and IV in green; ATP synthase is marked in red. Q, ubiquinone; Cyt C, cytochrome c.

**Figure 2 cells-11-01598-f002:**
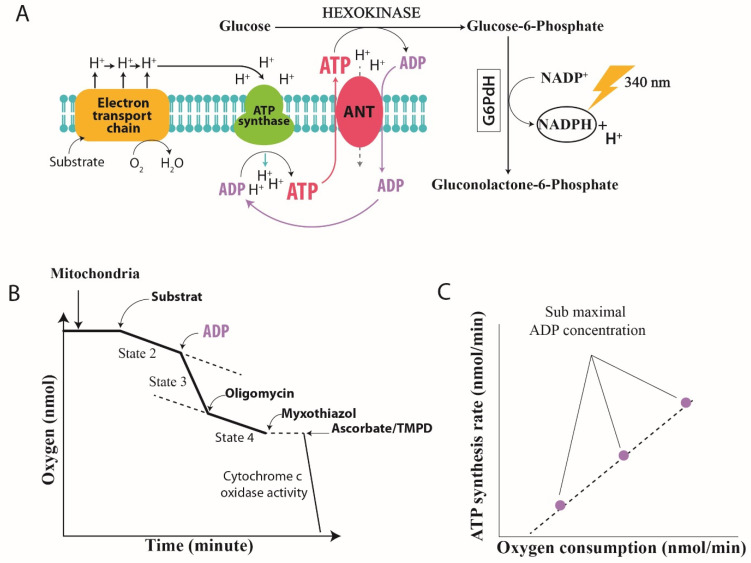
Assay chemistry and instrumentation. (**A**) The ATP:O ratio represents the ratio of ATP synthesis based on oxygen consumption measured simultaneously. It allows us to measure the efficiency of the oxidative phosphorylation. This ratio is determined using the “hexokinase system” based on the regeneration of ADP [17,25], thus making it possible to obtain a stable State 3. The intensity of State 3 can be modulated by increasing concentrations of ADP, resulting in the kinetics of ATP synthesis being obtained as a function of the consumption of oxygen between State 4 (the absence of ATP synthesis) and State 3 (maximum ATP synthesis). The reaction catalyzed by hexokinase (HK) is as follows: synthesized ATP is measured indirectly by evaluating glucose-6-phosphate production during the reaction. Glucose-6-phosphate dehydrogenase (G6PDH) allows the formation of NADPH from glucose-6-phosphate: HK glucose + ATP gGlucose-6-phosphate + ADP. The quantity of NADPH formed during this reaction can then be measured with a spectrophotometer at a wavelength of 340 nm and is directly proportional to the amount of ATP synthesized. (**B**) Typical trace of mitochondrial respiration. The respiration rate (determined by the slope of the curve) is calculated by the amount of consumed oxygen/minute/mg of protein. The State 2 non-phosphorylating state is respiration in the absence of ATP synthesis. State 3 is respiration in the presence of ATP synthesis. State 4 is the respiration in the presence of oligomycin, an ATP synthase inhibitor. Cytochrome c oxidase activity is the maximal respiration with ascorbate and N,N,N′N′-tetra methyl-p-phenylenediamine (TMPD) as a substrate. (**C**) ATP:O is the linear relationship between ATP production and oxygen consumption measured with the addition of different amounts of exogenous ADP. The mitochondrial ATP synthesis is initiated by the addition of different amounts of ADP. The intensity of State 3 can be modulated by increasing the concentration of ADP, resulting in the kinetics of ATP synthesis being obtained as a function of oxygen consumption between State 4 (absence of ATP synthesis) and State 3 (maximum ATP synthesis). There is a linear relationship between ATP synthesis and oxygen consumption when the oxidative phosphorylation rate increases by the addition of higher amounts of ADP.

**Figure 3 cells-11-01598-f003:**
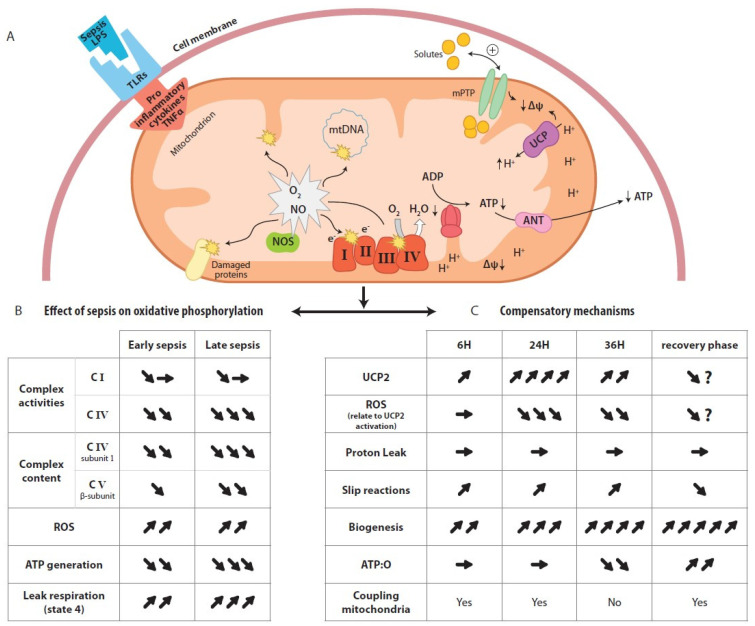
Effect of sepsis on mitochondrial oxidative phosphorylation and compensatory mechanisms for restoring oxidative phosphorylation efficiency (**A**) Interaction between bacterial lipopolysaccharide (LPS) and toll-like receptors (TLRs) induces pro inflammatory cytokines and triggers radical oxygen species (ROS) and nitric oxide (NO) generation. Increased in ROS and NO production cause damage to ETC components and ATP synthase, and induces a loss of membrane potential through the activation of mitochondrial permeability transition pores (mPTP). Activation of UCP 2 dissipates ROS production but it is not strong enough to prevent the uncoupling of oxidative phosphorylation. (**B**) Predicted consequences of damage to ETC components and ATP synthase in mitochondrial respiration, and ATP generation and ROS production across the time course of sepsis. (**C**) Time course of the response of mitochondrial bioenergetic and biogenesis parameters to sepsis, and the expected changes in the coupling efficiency in the recovery phase of sepsis, according to the current knowledge. Arrow (
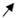
) indicate increase, arrow (
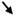
) indicate decrease, arrow (
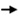
) indicate no changes.

## Data Availability

The study did not report any data.

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
