# Peer review of "Regulation of Oxidative Phosphorylation of Liver Mitochondria in Sepsis"

_cells, 2022, doi:10.3390/cells11101598_

Round 1

Reviewer 1 Report

Dear Authors,

The present manuscript is an interesting review of mitochondrial function and sepsis.

The main thing that I have some doubts it is your title and your conclusion. The way that you conduct your paper, I don't understand why you add in your title "role of cytochrome c oxidase". The paper did not focus in cyto C oxidase.

Another thing that I would suggest to improve your manuscript, is the quality of the image. In the pdf that I received, the quality of the image wasn't the best one for a paper in a journal with an IF = 6.6

Author Response

Dear reviewer

We thank you for your comments

We have removed role of cytochrome c oxidase from our title and

we have rework the figures, and removed the unnecessary colors .

we have attached the list of changes in the manuscript and the letter to the editor

Reviewer 2 Report

In the present article, the authors discussed the role of mitochondrial coupling loss and its corresponding decrease in oxidative phosphorylation in sepsis-induced liver damage. The authors attribute the decrease in mitochondrial coupling mainly to the slipping of respiratory complex IV and the decrease in cytochrome c oxidase level.

Although the topic is very interesting and the discussion about bioenergetics parameters and the methods description is well elaborate. The article puts too much weight on the possible cytochrome c oxidase slipping as the main responsible for the decrease in the production of ATP. This approach generates that the article barely discusses another very important factor like post-translational modifications induced by ROS, which are also capable of decreasing the activity of mitochondrial complexes as well as increasing mitochondrial membrane permeability, inducing loss of mitochondrial coupling. These modifications play are very important factors in a context of oxidative stress and inflammation characteristic of sepsis.

In the same sense, structural alterations induced in the mitochondrial cristae as a result of changes in the formation of ATP synthase dimers are not discussed in this article. These are of particular interest since the efficiency of ATP production is also directly dependent on them.

Furthermore, although UCPs are basic components of the intrinsic mitochondria decoupling systems, other recently was described that proteins such as adenine nucleotide translocator also contribute to the leak, so their discussion must be taken into account. Therefore, a deeper discussion of the aforementioned topics is necessary to enrich the manuscript before publication.

Minor Points.

  • Please check the order of the figures, figure 3 appears in the manuscript before figure 2
  • Integrative schemes of the mechanisms discussed could be very useful to enrich the manuscript

Author Response

Dear Reviewer

Thank you for your review.

Below the response of each of your remark and advice.

Major point

Comment 1

According to the reviewer’s comment, we have added other mechanisms that can explain a decreased in ATP generation. As suggested; we have mentioned alternative mechanisms such as cell signaling, the overall depletion of mitochondrial proteins consecutive to oxidation and nitration.

However, regarding our data on proton conductance and membrane potential, we showed that the loss of membrane potential is unlikely to be involved in a decrease of ATP generation in liver mitochondria during sepsis. We studied ATP/O with complex 1 and complex 2 substrates and  found no difference in oxidative phosphorylation efficiency.

Moreover, the consistent inhibition of cytochrome c oxidase in clinical  and experimental studies using the CLP model of sepsis has shown that cytochrome c oxidase and ATP synthase could  be a relevant therapeutic target, as changes in their stoichiometry could be an early response to sepsis because of the delay of synthesis ETC component. This explained our interest and particular focus on  this ETC component.

Comment 2 according to the reviewer’s comment we have mentioned structural alteration that can be involved in ATP depletion. We have also associated structural alteration with the depletion of ATP synthase subunits which could explain the disorganization of ATP synthase the increased in H/ATP stoichiometry .

Comment 3

According to referee’s comment we have discussed ANT and the ATP/ADP ratio as potential regulator of ATP generation efficiency. It appears that ANT is depleted in sepsis, so it is unlikely that it contributes significantly to dissipation of proton gradient in liver mitochondria. However UCP seems to be involve in increasing in a leak respiration as discussed.

Minor point

Integrative scheme summarized figure 3 and 4 have been added

The figure 2 and 3 now appear in the right order.

Note that we attached the letter to the editor as well as the changed made in the manuscript to the reply.

regards

Pierre Eyenga

Reviewer 3 Report

In this review the authors discuss previous findings regarding the regulatory mechanism involved in changes in liver mitochondria oxidative phosphorylation in sepsis with particular focus on the role of cytochrome c oxidase and much more briefly the therapeutics that could potentially improve cytochrome c oxidase function in sepsis.

I found some substantial criticisms in the review that in my opinion requires a major revision.

Major points

I consider that it is inappropriate and not necessary that two figures have been included in the review (figure 3 and figure 4) that report the experimental results and are already present in a work published by the same authors (Eyenga et al.2018 (ref 12).

I would eliminate both figures by replacing them with a new figure that schematizes the results.

Note that in the current version figure 3 has been inserted before figure 2.

Regarding Figure 2, the scheme shown in figure 2A reporting the spectrophotometric measurement of mitochondrial production of ATP erroneously shows NADH instead of NADPH which is the product of the reaction of G6PDH which has as a substrate G6P and as cofactor externally added NADP+ and not NAD+.

The legend relating to figure 2A was also not written correctly.

Lines 128-132: Describing the “approach to directly quantify mitochondrial ATP… by measuring ATP production spectrophotometrically in a system using hexokinase (HK) and glucose-6-phosphate dehydrogenase (G6-PDH) to convert ATP to nicotinamide adenine dinucleotide phosphate (NADPH)” the authors fail to cite the work of Passarella et al., 1988 Biochem. Biophys. Res. Commun. 156, 978–986, that firstly have developed a method to continuously monitor the ATP production which occurs by adding ADP to isolated rat liver mitochondria incubated in the presence of glucose, NADP+, hexokinase (HK), and glucose-6-phosphate dehydrogenase (G6PDH) by revealing the appearance of ATP in the extramitochondrial phase photometrically as an increase of absorbance at 340 nm due to NADP+ reduction to NADPH, taking into account that there is a stoichiometric ratio of 1:1 between ATP produced and formed NADPH.

The section on Therapeutics implication is too brief and poorly detailed and discussed

Conclusions are unclear regarding the role of cytochrome c oxidase and ATP synthase in regulating hepatic mitochondrial oxidative phosphorylation during sepsis

Minor Point

lines 80-81: Correct the oxidized and reduced forms of both nicotinamide and flavin coenzymes are the opposite of that reported

Line 140: Herminghaus et al., are ref (18) not (17)

Line 278: correct redox “slipping” instead of redox “sleeping”

Author Response

Dear Reviewer

Thank you for your review.

Below the response of each of your remark and advice.

Major point

Comment 1

According to the reviewer’s comment we have added a figures summarizing the figure 3 and 4 and removed the former figures 3 and 4 from the manuscript.

Comment 2

According to the reviewer comment, NADH have been changed in NADPH and NADP+. The legend of the figure has also been corrected

Comment 3

As requested, we have included the reference  Passarella et al. (1988)  (reference 17 ).

Comment 4

We have modified the ‘therapeutic section’ of our discussion. In particular, we have discussed the therapeutics which interact with the entire respiratory chain, cell signaling, mitochondrial biogenesis.

Comment 5

According to reviewer’s comment we have discussed the changes  stoichiometry of cytochrome c oxidase and ATP synthase as an early mitochondrial response to sepsis.

Minor point

Comment 1

Integrative schemes summarized figure 3 and 4 have been added and appeared now as a Figure 3.

The figure appears now in the right order.

Comment 1

We have corrected the oxidized  and reduced form of nicotinamide and flavin co enzymes

Comment 2

The Herminghaus reference have been changed

You will find attached to this reply the letter to the editor as well as the changes made in the manuscript

Regards

Pierre Eyenga

Round 2

Reviewer 3 Report

The manuscript has been extensively improved and modified by addressing adequately  to the reviewer's requests and can be accepted in its current form